# Functional Redundancy of *FLOWERING LOCUS T 3b* in Soybean Flowering Time Regulation

**DOI:** 10.3390/ijms23052497

**Published:** 2022-02-24

**Authors:** Qiang Su, Li Chen, Yupeng Cai, Yingying Chen, Shan Yuan, Min Li, Jialing Zhang, Shi Sun, Tianfu Han, Wensheng Hou

**Affiliations:** 1National Center for Transgenic Research in Plants, Institute of Crop Sciences, Chinese Academy of Agricultural Sciences, Beijing 100081, China; suqiangcaas@163.com (Q.S.); chenli01@caas.cn (L.C.); caiyupeng@caas.cn (Y.C.); 15846122179@163.com (Y.C.); l503986256@163.com (J.Z.); 2Ministry of Agriculture Key Laboratory of Soybean Biology (Beijing), Institute of Crop Sciences, Chinese Academy of Agricultural Sciences, Beijing 100081, China; yuanshan@caas.cn (S.Y.); limin0606edu@163.com (M.L.); sunshi@caas.cn (S.S.); hantianfu@caas.cn (T.H.)

**Keywords:** soybean, *GmFT3b*, flowering time, photoperiod, functional redundancy

## Abstract

Photoperiodic flowering is an important agronomic trait that determines adaptability and yield in soybean and is strongly influenced by *FLOWERING LOCUS T* (*FT*) genes. Due to the presence of multiple *FT* homologs in the genome, their functions in soybean are not fully understood. Here, we show that *GmFT3b* exhibits functional redundancy in regulating soybean photoperiodic flowering. Bioinformatic analysis revealed that *GmFT3b* is a typical floral inducer *FT* homolog and that the protein is localized to the nucleus. Moreover, *GmFT3b* expression was induced by photoperiod and circadian rhythm and was more responsive to long-day (LD) conditions. We generated a homozygous *ft3b* knockout and three *GmFT3b*-overexpressing soybean lines for evaluation under different photoperiods. There were no significant differences in flowering time between the wild-type, the *GmFT3b* overexpressors, and the *ft3b* knockouts under natural long-day, short-day, or LD conditions. Although the downstream flowering-related genes *GmFUL1* (*a, b*), *GmAP1d*, and *GmLFY1* were slightly down-regulated in *ft3b* plants, the floral inducers *GmFT5a* and *GmFT5b* were highly expressed, indicating potential compensation for the loss of *GmFT3b*. We suggest that *GmFT3b* acts redundantly in flowering time regulation and may be compensated by other *FT* homologs in soybean.

## 1. Introduction

The change from vegetative to reproductive growth is a critical developmental transition in the life of flowering plants. Time to flowering directly influences crop maturity and determines adaptability to diverse geographic regions [1]. Proper flowering time is a prerequisite for soybean yield, and a series of studies have focused on optimizing soybean flowering time to maintain productivity during introduction to new regions [1,2,3].

In *Arabidopsis thaliana*, a long-day (LD) plant, flowering is induced by external and endogenous cues such as photoperiod, gibberellin levels, vernalization, and autonomous flowering signaling [4]. Among these cues, photoperiodical variation directly affects not only flowering time but also the podding stage and time to maturity [5]. Florigen is a compound produced in leaves and transmitted to the shoot apical meristem (SAM) to initiate flowering [6,7]. In *Arabidopsis*, florigen is the key regulatory integration factor in flowering induction pathways [7]. Recent research has shown that FLOWERING LOCUS T (FT) homologs (FTs), a family of phosphatidylethanolamine-binding proteins (PEBPs), have florigen function [6,7]. Several photoperiodical regulatory pathways determine flowering: *GIGANTEA* (*GI*), *CONSTANS* (*CO*), and *FT* function as central components in triggering flowering under LD conditions [7,8,9]. In brief, the circadian clock GI activates *CO* expression by binding to its promoter under LD photoperiod conditions, but not under short-day (SD) photoperiod conditions [10]. Accumulated CO protein directly activates the expression of *FT* in leaves [9,10]. Furthermore, FT moves to the SAM, where it forms a complex by interacting with FLOWERING LOCUS D (FD) [11,12]. The FT-FD complex induces expression of flowering-related genes, such as *SUPPRESSOR OF OVEREXPRESSION OF CO 1* (*SOC1*), *FRUITFULL* (*FUL*), and *LEAFY* (*LFY*), and finally initiates the floral meristem identity gene *APETALA1* (*AP1*) to stimulate flowering [11,13,14,15,16]. Likewise, in rice, the FT homolog protein Heading date 3a (Hd3a) interacts with the FD homolog OsFD1 in the SAM, assisted by 14-3-3 proteins [17]. These complexes then bind to the promoter of *OsMADS15* (an *AP1* homolog) to activate floral transition in rice, which highlights conservation of the flowering regulatory module (FT/FD-AP1) between different photoperiodic plants [17,18].

Soybean, a typical short-day (SD) plant, is particularly sensitive to photoperiods and is considered a classical photoperiodic model plant [5]. Day length determines plant flowering time; for example, in soybean, SD accelerates flowering, whereas LD represses flower bud formation [19,20]. Soybeans can be cultivated at latitudes ranging from ~20° N in the south to ~50° N in the north [21]. Unfortunately, most soybean varieties have limited latitude adaptability, resulting in a narrow cultivation area for each variety [5]. Flowering time and maturity period are the key agronomic traits that directly determine the yield and quality of soybeans [5,19]. Thus, an in-depth understanding of the roles of photoperiod genes at the molecular level is of great significance for adaptation of soybean varieties to diverse geographic regions.

To date, several key genetic loci have been identified that have large effects on flowering time and maturity period in soybean. These include *E1*-*E11*, *J*, *Tof11*, and *Tof12*, which comprise the phytochrome–clock-related gene *E1*-*FTs* flowering pathway [3,5,20,22,23,24]. *FTs* redundantly control photoperiod-regulated flowering in soybean. Specifically, *GmFT2a* (*Glyma16g26660*) and *GmFT5a* (*Glyma16g04830*) have been proven to effectively promote flowering through photoperiod regulation [20,25]. Interestingly, *GmFT2a* has a stronger effect on floral initiation under SD conditions, whereas *GmFT5a* has a stronger effect on flower induction under LD conditions [25]. GmFT2a and GmFT5a interact with both GmFDL12 and GmFDL19, but only GmFT5a interacts with GmFDL06 [26]. This may cause functional differentiation of *FT* genes. In addition, *GmFT2b* (*Glyma16g26690*) promotes flowering under LD [2]. *GmFT5b* (*Glyma19g28400*) promotes early flowering in *Arabidopsis* [27]. In contrast, *GmFT1a* (*Glyma18g53680*) was shown to be upregulated by *E1* and to delay flowering and maturity, confirming the identity of flowering inhibitors [28]. Similarly, *GmFT4* (*Glyma08g47810*) also delays flowering in *Arabidopsis* plants and may be the relevant gene in the *E10* locus [29,30]. Duplication and divergence of ancestral *FT* genes have produced multiple flowering regulatory proteins, some of which have antagonistic functions in flowering in sugar beet [31], apple [32], and onion [33]. This demonstrates that *FTs* have undergone diversified functional changes during the evolution of various crops and that photoperiod-dependent flowering is strictly controlled by coordinated expression of *FT* family genes.

Soybean *FTs* have been evaluated in various species. For example, ectopic expression of *GmFT3b* induces early flowering in *Arabidopsis* [27]. In this study, we found that *GmFT3b* redundantly participated in soybean photoperiodic flowering. First, *GmFT3b* expression was confirmed to be photoperiod-dependent and more responsive to LD conditions. We generated *GmFT3b*-overexpressing soybean plants and *ft3b* knockout plants and evaluated them under various photoperiods. Based on the expression profiles of flowering-related genes, we here propose a model where *FTs* redundantly regulate flowering in soybean.

## 2. Results

### 2.1. Identification of the FT Homolog GmFT3b in Soybean

According to the *Glyma19g28390* gene sequence in the Phytozome database, *GmFT3b* was cloned from the soybean variety Jack and sequenced. The 2336 bp genomic sequence of *GmFT3b* contained four introns and three exons (Appendix A), including an open reading frame (ORF) 546 bp in length. The ORF encoded a product 175 aa residues in length with a molecular weight of 19.72 kDa.

Some *FTs* have developed opposing functions during evolution, antagonizing plant flowering processes in *Arabidopsis* [27], soybean [28], and tobacco [34]. Thus, we performed multiple sequence alignment of GmFT3b with other FTs that have been well characterized in multiple species [35]. The results show that GmFT3b belongs to the *FT* family, with a highly conserved PEBP domain from 32 aa to 162 aa. GmFT3b also contains a tyrosine residue at position 134, which is consistent with all flowering inducer FTs except GmFT5a. In contrast, repressor FTs (AtFTL1, GmFT4, and NtFT1) are not tyrosine residue at position 134 (Figure 1A,B).

Phylogenetic analysis was performed with GmFT3b and selected PEBP proteins from soybean, *Arabidopsis*, *Beta vulgaris*, and *Malus domestica*. Both GmFT3b and GmFT3a clustered with inducer FTs such as GmFT2a, GmFT2b, and *Arabidopsis* FT (Figure 1C). GmFT3b therefore appears evolutionarily conserved and closely related to soybean flowering inducer GmFT2a, suggesting that GmFT3b contains conserved elements that may positively regulate flowering.

### 2.2. GmFT3b Is Localized to the Nucleus

Previous studies showed that the FT protein was localized to the nucleus and acted as an integration factor in soybean [2,28]. We generated a construct containing a *GmFT3b-GFP* fusion gene driven by the 35S-CaMV promoter (the PTF101-GFP-GmFT3b vector), then assessed the subcellular localization of GmFT3b-GFP in tobacco leaves with transient expression of the plasmid. As expected, GmFT3b-GFP was mainly expressed in the nucleus, as demonstrated by colocalization with the red nuclear marker fluorescent fusion protein NM-mCherry (Figure 2A). In addition, immunoblot analysis of total nuclear proteins confirmed that GmFT3b-GFP/GFP were expressed in the nucleus of tobacco leaves as expected (Figure 2B,C).

### 2.3. Day Length and Circadian Rhythm Regulate the Expression Pattern of GmFT3b

*FT* functions as a florigen to induce floral transition, and its expression patterns are regulated by photoperiod and circadian rhythm [20,28]. Previously, we evaluated two varieties with extreme photoperiod response phenotypes, the early-flowering variety Heihe 27 (HH27) and the late-flowering variety ZiGongDongDou (ZGDD). Here, the diurnal expression patterns of *GmFT3b* were analyzed in leaves of HH27 and ZGDD plants under various photoperiodic conditions. In both varieties, *GmFT3b* showed diurnal circadian rhythm under SD and LD conditions (Figure 3A,B). Under SD conditions, *GmFT3b* remained highly expressed in both varieties and peaked at 4 h after dark in HH27, but not in ZGDD plants. *GmFT3b* expression patterns were also comparable to one another in HH27 and ZGDD under LD conditions, although *GmFT3b* levels peaked 2 h earlier in ZGDD than in HH27. The results suggested that *GmFT3b* expression was regulated by circadian rhythm and was more sensitive to the induced LD photoperiod compared to the SD.

### 2.4. Evaluation of GmFT3b-Overexpressing Soybean Plants under Different Photoperiods

To investigate the effects of *GmFT3b* in plant flowering, we created three *GmFT3b*-overexpressing soybean lines (named OE3, OE5, OE6) via *Agrobacterium*-mediated transformation. The *GmFT3b*-overexpressing plants were identified via PCR and LibertyLink strips (Figure 4A,B). Western blot analysis indicated that the GFP-GmFT3b fusion protein was expressed in the T_2_ generations of OE3, OE5, and OE6 plants, but not in WT plants (Figure 4C). These results indicated that *GmFT3b* was inserted into the soybean genome and successfully translated.

The T_2_ generations of *GmFT3b*-overexpressing plants were planted along with WT under natural long-day (NLD), SD, and LD conditions; first flower appearance time was recorded in days after emergence (DAE). Under NLD conditions, the WT plants flowered at 27.8 d, and there was no significant difference compared to the *GmFT3b*-overexpressing plants (OE3, 27.8 d; OE5, 28.5 d; OE6, 28.9 d) (Figure 4D,E). Flowering time under SD conditions was not significantly different between WT (22.7 d) and the *GmFT3b* overexpressors (OE3, 22.6 d; OE5, 22.5 d; OE6, 21.8 d) (Figure 4F). There were also no significant differences in flowering time under LD conditions (WT, 42.3 d; OE3, 42.3 d; OE5, 41.5 d; OE6, 42.8 d) (Figure 4G).

### 2.5. ft3b Knockout Did Not Affect Flowering Time

To further investigate the function of *GmFT3b* in soybean, we used CRISPR/Cas9 to generate an *ft3b* knockout soybean line. First, the genome editing target was placed near the start codon of the first exon of *GmFT3b* (Figure 5A). The genomic target sequence of *GmFT3b* near the cleavage site was amplified and confirmed by sequencing. Finally, we obtained a homozygous *ft3b* mutant with a 72 bp deletion that resulted in a missing start codon, preventing normal translation (Figure 5B,C).

Flowering times of the *Gmft3b* knockouts were evaluated under different photoperiods. Under NLD conditions, *Gmft3b* mutant plants flowered at 28.3 d, which was not significantly different compared to WT plants (27.8 d) (Figure 5D,E). There were also no significant differences in flowering time under SD conditions (*Gmft3b*, 22.4 d; WT, 22.7 d) (Figure 5F) or LD conditions (*Gmft3b,* 41.2 d; WT, 42.3 d) (Figure 5G). These results indicated that mutation of *GmFT3b* alone did not alter soybean flowering performance under several photoperiodic environments.

### 2.6. Expression of Downstream Flowering-Related Genes

It was previously reported that *FT* positively regulates expression of *SOC1*, *FUL*, and *LFY* homologs in the SAM of *Arabidopsis* and soybean [13,14,15,25]. We therefore used quantitative reverse-transcription PCR (qRT-PCR) to assess the expression profiles of *FTs* and downstream flowering-related genes. Under NLD photoperiod, as expected, *GmFT3b* expression was significantly higher in *GmFT3b* overexpressors and lower in *ft3b* knockouts compared with WT plants (Figure 6). Expression levels of *GmAP1d, GmLFY1*, and *GmFUL1* (*a*, *b*) were significantly lower in *ft3b* mutant plants but not in *GmFT3b* overexpressors, with the exception of *FUL1b* (Figure 6).

The genetic compensation response (GCR) mechanism can significantly increase expression of other *FT**s* in single *ft* soybean mutants [1]. We therefore also measured expression of *FT**s* in the *FT3b* knockouts and overexpressors. In *GmFT3b*-overexpressing soybean plants, no significant differences in *FT* levels were observed compared with WT plants. As expected, *GmFT5a* and *GmFT5b*, two flowering inducers, were upregulated in *ft3b* mutant plants. Taken together, the results showed that neither increasing nor decreasing *GmFT3b* expression levels affected expression of downstream flowering-related genes in soybean.

## 3. Discussion

Soybean provides more than a quarter of the total protein consumed by humans and animals worldwide, meaning that soybean directly affects the quality of human life. Cultivated soybeans are paleo tetraploids that were domesticated from wild soybean (*Glycine soja* Sieb. et Zucc.) ~5000 years ago [3,36]. Thus, the soybean genome is complex, and ~75% of protein-coding genes have multiple copies [37]. For example, more than 11 *FTs* are present in the soybean genome [19,20,38]. However, most studies have focused on only two of these, *GmFT2a* and *GmFT5a* [25,26,39]. The other *FTs* were discovered more than ten years ago, at which time studies were limited in *Arabidopsis* [27]. Therefore, in-depth research to fully explain the function of other *FTs* (such as *GmFT3b*) is necessary to facilitate the understanding of the photoperiodic flowering pathways in soybean.

After gene duplication events, the redundant genes can undergo functional conservation, neofunctionalization, subfunctionalization, or functional degradation during evolution [40,41]. As a result, some *FT* homologs display opposing biochemical actions [28,29,32,33,35]. *GmFT3b* is a member of the PEBP homologs and shows high sequence similarity with flower-inducing *FTs* (Figure 1A,B). The GmFT3b-GFP fusion protein was primarily observed in the nucleus (Figure 2A–C), indicating that GmFT3b was localized to the nucleus with other FT proteins [2,28]. Photoperiod and circadian rhythm regulate the overall transcription and diurnal expression patterns of *FT* genes, thereby systematically and accurately controlling the transition from vegetative to reproductive growth [20,28]. Three expression patterns of *FT* were preliminarily identified in cultivated soybean varieties, and GmFT3b belonged to the photoperiod-independent group [28]. However, *GmFT3b* was determined in the present study to be photoperiod-dependent, with expression patterns depending on day length and changing more strongly under LD than SD conditions (Figure 3A,B). In addition, considering that *GmFT3b* promotes flowering in *Arabidopsis* [27], *GmFT3b* might retain the function of controlling flowering in soybean.

Several recent studies showed that *GmFT2a* and *GmFT5a* are functionally equivalent to the *Arabidopsis FT* that induce early flowering, with both able to rescue the *Arabidopsis ft-10* mutant [20,37]. Likewise, Lee et al. found that *GmFT3b* acted as a strong flowering inducer in *Arabidopsis* [27]. In contrast, *GmFT3b* expression had a negligible influence on flowering in soybean; there were no significant differences in floral transition time between WT, three *GmFT3b*-overexpressing soybean lines, and an *ft3b* knockout line under NLD, SD, or LD conditions. Interestingly, soybean plants overexpressing *GmFT5a* failed to induce early flowering under SD conditions because only *GmFT2a* also promoted flowering [25]. Similarly, we speculated that the expression levels of *GmFT2a*, *GmFT5a* and *GmFT5b* were sufficient to induce flowering under LD photoperiods, although the *GmFT3b* was overexpressed in soybean. It would intuitively be expected that knocking out *FT* genes would drastically affect downstream flowering-related genes [1,25], but the *ft3b* knockouts did not show altered expression of *AP1* (*a*-*c*), *LFY2*, *FULa*, or *SOC1* (*a*, *b*) compared with the WT (Figure 6). GCR is widespread in animals and plants, which leads to weaker phenotypes in single mutants [1,42]. Strikingly, *Gmft2a* and *Gmft5a* single mutants displayed weak roles in activating flowering compared with the double mutants, while the expression levels of other *FT* genes were increased [1]. Similarly, *GmFT5a* and *GmFT5b* were upregulated in *ft3b* plants in this study, suggesting *GmFT5a* and *GmFT5b* compensate for the function of *GmFT3b*. In particular, the double mutations of *ft3b* with *Gmft5a* or *Gmft5b* should be created to further investigate the function of *GmFT3b* in future studies. Taken together, the results suggested that a loss-of-function mutation in *GmFT3b* was compensated by other *FT* genes in photoperiodic flowering.

Short, synchronous flowering time is considered a critical trait for crop yield. Interestingly, the domesticated *FT* alleles slightly delay flowering compared with wild alleles, which may be a result of losing *FT* duplicates [37,43]. Moreover, loss of function in *FT*s by natural or artificial mutation clearly determined the ecological adaptation range of soybean [1,20,25]. Thus, variation in *FT*s is an important basis for diversity in flowering time and maturity, which contribute to soybean geographical adaptability. We previously showed that *GmFT3b* may have undergone breeding selection, and its haplotypes were associated with flowering time and maturity [44]. However, *GmFT3b* was functionally redundant in regulation of flowering time under our experimental conditions. In addition, almost all identified polymorphisms were distributed in gene regulatory regions (the 5′ UTR, 3′ UTR, and intron regions) [44], implying that differences in *GmFT3b* between soybean varieties are at the transcriptional level. Therefore, it is necessary to study the function of *GmFT3b* in different soybean backgrounds in future studies. Besides, environmental stresses including drought, salinity, heat, and nutrient stress can also affect the flowering [45]. Although we explored the function of *GmFT3b* under different photoperiods, it would be interesting to study the role of *GmFT3b* under environmental stress in the future. Taken together, we propose a model where *FTs* redundantly regulate flowering in soybean (Figure 7). Under LD photoperiod conditions, the light signal is received by photoreceptors (*E3* and *E4*), then transmitted to *FTs* through a photoperiod-dependent pathway [3,7,19,20]. *GmFT5a*, *GmFT2a*, *GmFT5b*, and *GmFT3b* act as floral activators, with *GmFT5a* having a decisive influence and the ability to compensate for the function of *GmFT3b*. Floral activators are required to counteract flowering inhibitors to activate downstream flowering-related gene expression and subsequently induce flowering. Further studies are needed to determine how *GmFT3b* is compensated in soybean.

## 4. Materials and Methods

### 4.1. Plant Materials and Growth Conditions

Soybean variety Jack was used in this study. The seeds of Jack, *GmFT3b*-overexpressing, and *ft3b* mutants were sown in plastic pots, which were placed in standard long-day (16 h light/8 h dark, 22–30 °C) or short-day (12 h light/12 h dark, 22–30 °C) growth rooms (PPFD, 299.72 μmol/m^2^s; CCT, 3190 K; Lux, 11068 lx), respectively. Besides, materials were planted under natural long-day conditions in Beijing (116°33′ E, 39°96′ N; 15 May–30 September 2021).

### 4.2. Bioinformatics Analysis

The amino acid sequence of *FT* and *TFL* homologs was retrieved from NCBI (https://www.ncbi.nlm.nih.gov/; accessed on 19 May 2021) and Phytozome (https://phytozome.jgi.doe.gov/pz/portal.html/; accessed on 19 May 2021) database. The multiple sequence alignment was performed using the Clustal Omega (https://www.ebi.ac.uk/Tools/msa/clustalo/; accessed on 19 May 2021) web program. The phylogenetic tree was constructed using the maximum likelihood method by MEGA X software.

### 4.3. Generation of GmFT3b-Overexpressing Soybean Plants

The soybean variety was used for transformation according to the protocol reported previously [46]. To generate the *GmFT3b*-overexpressing soybean plants, the full-length *GmFT3b* sequence was cloned from the cDNA library of soybean shoot apex using GmFT3b-FQ-F/R primers and then subcloned to the overexpression vector PTF101-GFP with the GmFT3b-101F/R primers, named PTF101-GFP-GmFT3b. Subsequently, the constructed vector was directly transformed into *Agrobacterium tumefaciens* strain EHA101 and generated T_0_ generation of *GmFT3b*-overexpressing plant lines. Next, the T_1_ generations of *GmFT3b*-overexpressing seedlings were screened by glufosinate herbicide and confirmed by PCR using the FT3b-JC351F/R primers, and the T_2_ generations of *GmFT3b*-overexpressing plants were used in this study. All PCR primers were listed in Appendix B.

### 4.4. CRISPR-Mediated Mutation of GmFT3b

The sgRNA of *GmFT3b* (GmFT3b-TS: AGAGGGTTCCTACTACCGCCAGG) was selected by the CRISPR-P web server (http://cbi.hzau.edu.cn/cgi-bin/CRISPR; Accessed on 18 July 2018). Then, the oligo sequence of GmFT3b-TS was synthesized and inserted into the CRISPR/Cas9 vector that was driven by the *AtU6* promoter. Next, the CRISPR/Cas9 vector was transformed into the *A. tumefaciens* strain EHA105 and then introduced into soybean variety Jack through *Agrobacterium*-mediated transformation as described previously [46]. All T_0_
*GmFT3b*-overexpressing were screened by PCR with the detection primers (FT3b-439F/FT3b-439R) and subsequently confirmed by sequence. In addition, an *ft3b* homozygous mutant was detected by both PCR and Bar test strip methods as described previously [47]. All primers used in this study are listed in Appendix B.

### 4.5. Gene Expression Analysis

The shoot apex tissues of soybean were harvested at 20 DAE (natural long-day conditions), 14 DAE (short-day conditions), and 35 DAE (long-day conditions), respectively. According to the manufacturer’s instructions, total RNA was extracted by TRIzol reagent (Invitrogen, Carlsbad, CA, USA). Then, ~1 μg total RNA was reverse transcribed into cDNA via HiScript^®^ III RT SuperMix for qPCR kits (Vazyme, Nanjing, China). Quantitative real-time PCR (qRT-PCR) was performed on ABI QuantStudio 7 Flex (Applied Biosystems, San Francisco, CA, USA) with ChamQ SYBR qPCR Master Mix (Low ROX Premixed) (Vazyme, Nanjing, China). The qRT-PCR program followed the manufacturer’s instructions, and all samples tested in expression analysis were verified with three technical replications. The qRT-PCR data were determined by 2^−ΔΔCT^ methods [48], and the statistical significance of differences was analyzed by Microsoft Excel by using the one-way ANOVA method. All primers are listed in Appendix B.

### 4.6. Subcellular Localization of GmFT3b

The subcellular localization of GmFT3b was conducted in *Nicotiana benthamiana* plants using *Agrobacterium*-mediated transformation as described previously [49]. Briefly, *N. benthamiana* plants were grown in pots under long-day conditions (16 h/light, 8 h/dark; 22–30 °C). Then, the *N. benthamiana* leaves were injected with *A. tumefaciens* strain GV3101, which carried PTF101-GFP-GmFT3b/pTF101-GFP and nuclear marker plasmids together, respectively. After 48 h agroinfiltration, the *N. benthamiana* leaves were collected and imaged by FLUOVIEW FV3000 Confocal Laser Scanning Microscope (Olympus Corporation, Tokyo, Japan).

### 4.7. Western Blot Analysis

The first trifoliate leaf of *GmFT3b*-overexpressing seedlings was harvested and conserved under −80 °C. The total soluble proteins were extracted using the plant protein extraction protocol as described previously [50]. The total nuclear proteins were extracted using the nuclear and cytoplasmic extraction kit (CWBIO, Beijing, China). After protein denaturation, samples were separated by 10% SDS-PAGE and then analyzed by immunoblot using 1:3000-fold dilution anti-GFP mouse monoclonal antibody. In addition, the PVDF membrane was incubated with the One Step Western Kit HRP (CWBIO, Beijing, China). Finally, the immunized proteins were imaged under the Amersham Imager 600 (GE Healthcare, Little Chalfont, Buckinghamshire, UK) machine.

## 5. Conclusions

*GmFT3b* is a typical *FT* homolog. Soybean lines overexpressing *GmFT3b* and *ft3b* knockout soybean plants were used to investigate the effects of *GmFT3b* in regulation of photoperiodic flowering for the first time. Neither overexpression nor knockout of *GmFT3b* significantly affected flowering time or expression of downstream flowering-related genes. Based on these data, we suggest that other *FT* homologs are functionally redundant with *GmFT3b* in regulation of photoperiodic flowering and that the homologs may compensate for the loss of *GmFT3b*.

## Figures and Tables

**Figure 1 ijms-23-02497-f001:**
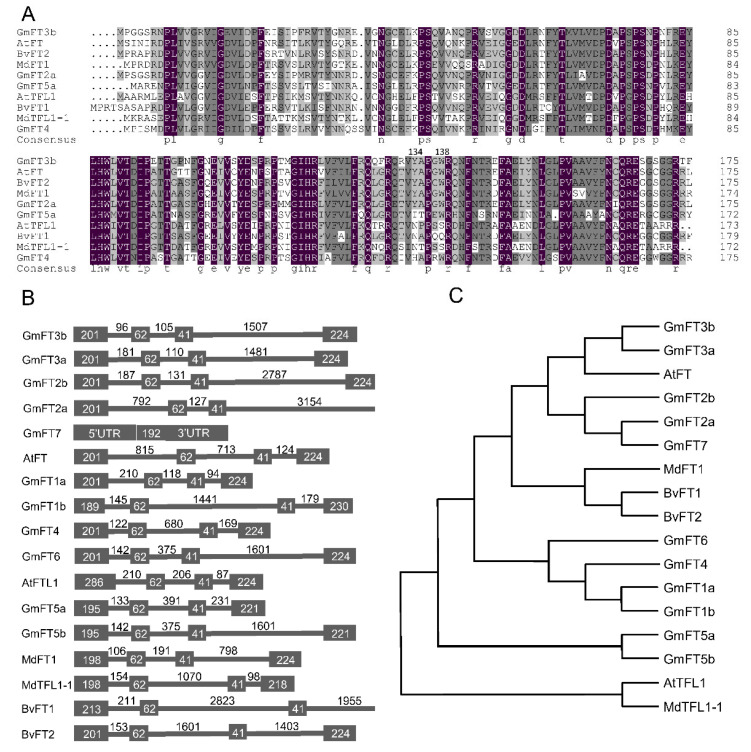
Identification of the GmFT3b and sequence analysis of PEBP family members. (**A**) Multiple sequence alignment is achieved by the Clustal Omega web program (http://www.clustal.org/omega/; accessed on 19 May 2021). (**B**) Genomic organization of the PEBP family members. The boxes and lines indicate exons and introns, respectively. (**C**) The software MEGA X was used to perform the phylogenetic tree analysis. Protein sequences were obtained from NCBI, Phytozome and TAIR database as follows: AT1G65480.1 (AtFT), AT5G03840.1 (AtTFL1), Glyma18g53680 (GmFT1a), Glyma18g53690 (GmFT1b), Glyma16g26660 (GmFT2a), Glyma16g26690 (GmFT2b), Glyma16g04840 (GmFT3a), Glyma19g28390 (GmFT3b), Glyma08g47810 (GmFT4), Glyma16g04830 (GmFT5a), Glyma19g28400 (GmFT5b), Glyma08g47820 (GmFT6), Glyma02g07650 (GmFT7), ADM92608.1 (BvFT1), ADM92610.1 (BvFT2), HQ424013.1 (MdFT1), AB162040.1 (MdTFL1-1).

**Figure 2 ijms-23-02497-f002:**
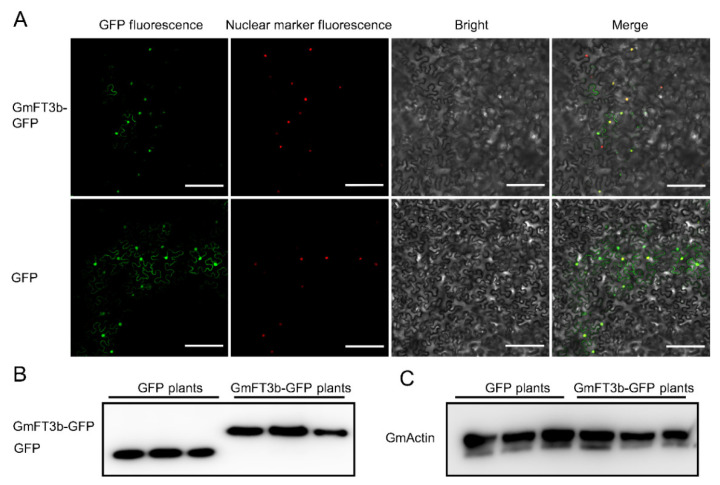
Colocalization and expression pattern analysis of *GmFT3b.* (**A**) Subcellular localization of GmFT3b in *Nicotiana benthamiana* leaves. The GmFT3b-GFP/GFP was co-transformed with nuclear marker (red) NM–mRFP into *N. benthamiana* leaves, and the GFP signal (green) completely colocalized with the nuclear signal (red). Scale bar, 20 μm. Immunoblot analysis of the transient expressed GmFT3b-GFP (**B**) and GmActin (**C**) in *N. benthamiana* leaves. The GmFT3b-GFP/GFP and GmActin were immunized by anti-GFP and anti-Actin polyclonal antibody, respectively.

**Figure 3 ijms-23-02497-f003:**
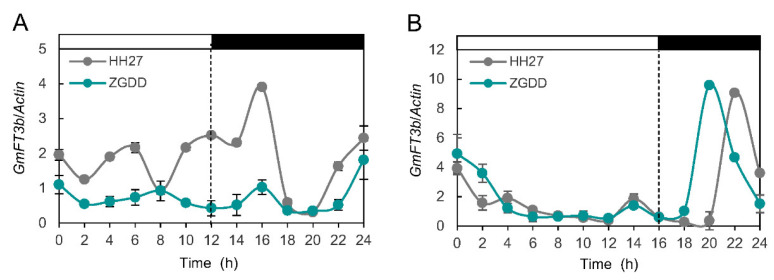
The daily expression patterns of *GmFT3b* under SD (**A**) and LD (**B**) conditions. Zigongdongdou (ZGDD) or Heihe27 (HH27) is late- or early-flowering soybean variety, respectively. The light and dark phases are represented by white and black bars, respectively.

**Figure 4 ijms-23-02497-f004:**
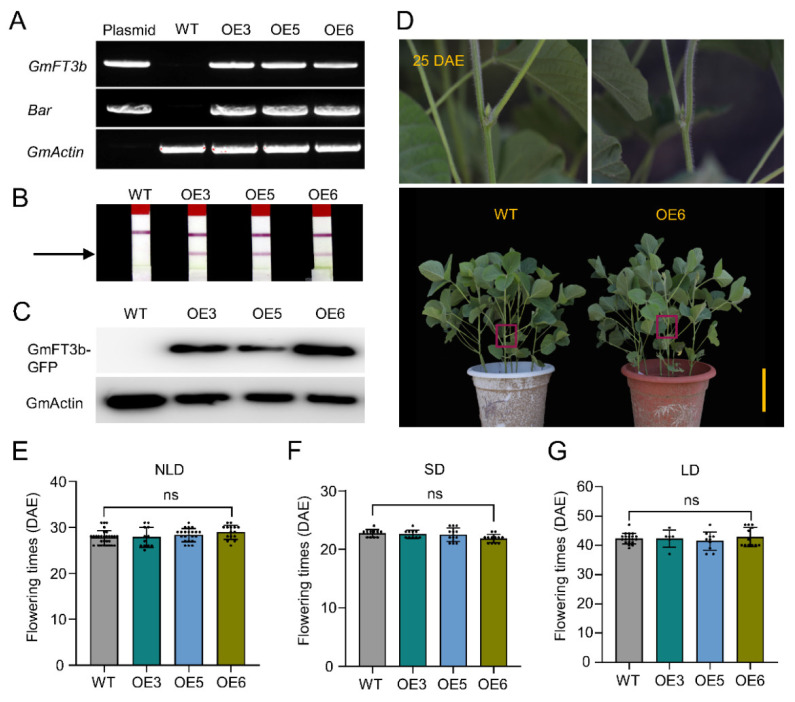
Identification and evaluation of *GmFT3b*-overexpressing soybean plants. The *GmFT3b*-overexpressing soybean plants were detected by PCR (**A**), LibertyLink strips (**B**), and Western blot (**C**). *GmActin* (Glyma08G146500) and *Bar* acted as internal reference genes for PCR. GmFT3b-GFP fusion protein was detected by anti-GFP antibody. WT represents the transformation recipient variety Jack. The arrow indicates Bar protein. The phenotypes of WT and *GmFT3b*-overexpressing soybean plants under NLD conditions (**D**). The flowering time of *GmFT3b*-overexpressing soybean plants under NLD (**E**), SD (**F**), and LD conditions (**G**). DAE, days after emergence. The dots indicate the plants used for counting the days to first flower appearance. ns indicates not significant. The significant differences are determined by one-way ANOVA. Error bars indicate standard deviation. Scale bar, 20 cm.

**Figure 5 ijms-23-02497-f005:**
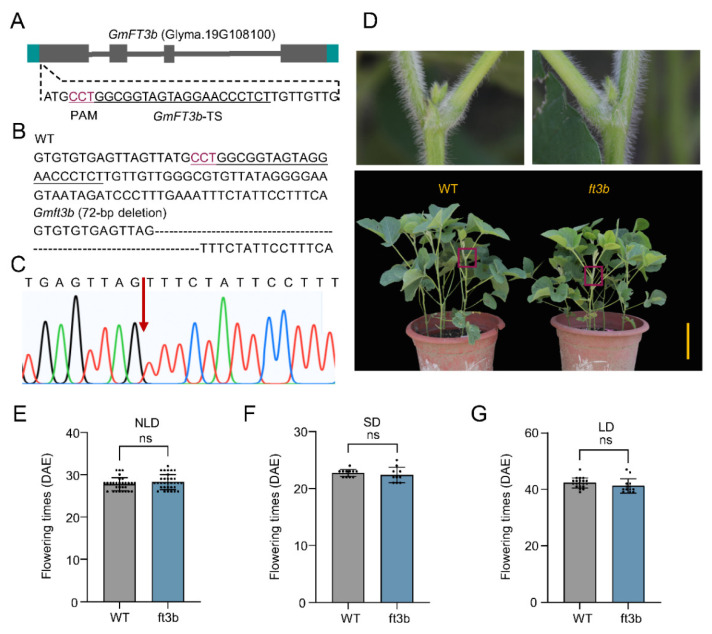
Homozygous-targeted mutagenesis of *GmFT3b* induced by CRISPR/Cas9. (**A**) The structure and target sites of *GmFT3b*. Underlined sequence indicates the target site of *GmFT3b.* The red sequence is the PAM region. (**B**) CRISPR/Cas9-induced mutations at the targeting sites. “–” indicates deletion of nucleotides. (**C**) Detailed sequence of the target site of *GmFT3b* in the *ft3b* plants. The arrow indicates the site of the base deletion. The phenotypes of WT and *ft3b* plants under NLD conditions (**D**). The flowering time of *ft3b* plants under NLD (**E**), SD (**F**), and LD conditions (**G**). DAE, days after emergence. The dots indicate the plants used for counting the days to first flower appearance. ns indicates not significant. The significant differences are determined by one-way ANOVA. Error bars indicate standard deviation. Scale bar, 20 cm.

**Figure 6 ijms-23-02497-f006:**
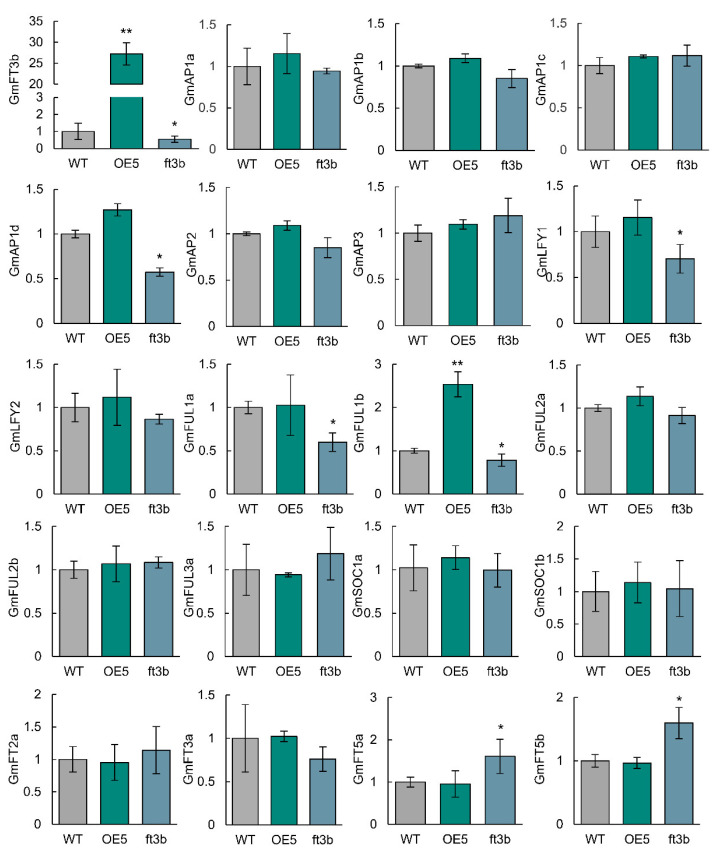
The expression levels of flowering-related genes and *FTs* in shoot apex of *GmFT3b* overexpressors, *ft3b* mutants, and WT plants under NLD conditions. Three technical replicates were analyzed in this experiment. Error bar indicates the SE values. Asterisks indicate that there are significant differences between *GmFT3b* overexpressors (*ft3b* mutants) and WT plants (*, *p* < 0.05; ****, *p* < 0.01, Student’s *t*-test).

**Figure 7 ijms-23-02497-f007:**
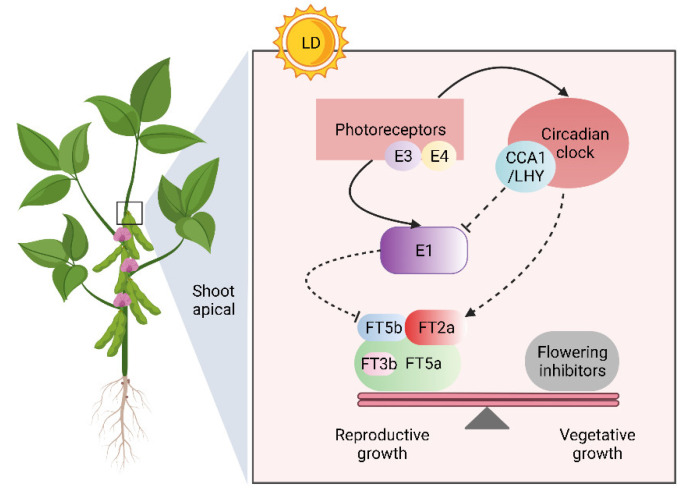
Schematic model depicting the *FTs* redundancy regulating flowering under LD condition. The *FTs* integrate the photoperiodic signal by photoperiod-dependent pathway. *GmFT5a*, *GmFT2a*, *GmFT5b*, and *GmFT3b* function as floral activators complex to redundantly transmit flowering information, and *GmFT5a* plays a decisive role in this complex. Floral activators overcome the effects of inhibitors to upregulate the downstream flowering-related genes and induce flowering. The sizes of boxes represent the effects of *FTs* in photoperiodic flowering. Arrow and bar-ended represent the promotion and inhibition effect, respectively. Dashed lines represent unverified regulatory mechanism.

## Data Availability

All data generated and analyzed in this study are included in this paper.

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
