# Peer review of "Functional Redundancy of FLOWERING LOCUS T 3b in Soybean Flowering Time Regulation"

_ijms, 2022, doi:10.3390/ijms23052497_

Round 1

Reviewer 1 Report

The manuscript entitled "Functional redundancy of FLOWERING LOCUS T 3b in soybean flowering time regulation" is interesting and the experiment has been done well.

The authors need to explain the possible reasons for the redundancy of GmFT3b because no effect on the flowering in either the overexpression or knockout mutants of this gene. This may mean redundancy but maybe not be functional in a plant.

Figure 7 shows the function of FT3b, but I cannot imagine how FT3b interacts with FT5b, FT2s, and GT5a. What is the function of FT3b in this complex to promote reproductive growth in soybean?

Author Response

Point 1: The authors need to explain the possible reasons for the redundancy of GmFT3b because no effect on the flowering in either the overexpression or knockout mutants of this gene. This may mean redundancy but maybe not be functional in a plant.

Response 1: Thank you for your careful and constructive comments. The soybean genome is complex, with at least 11 FT genes. Among them, GmFT2a and GmFT5a play decisive roles in controlling flowering. Recently, Li et al. (2021) found both GmFT2a and GmFT5a can be complemented by each other in single mutants, while their double mutants displayed enhanced phenotypes. In this study, the flowering inducers GmFT5a and GmFT5b were up-regulated in ft3b mutant plants. Therefore, we reasoned that, similar to the complementation mechanism of GmFT2a and GmFT5a, GmFT3b might be compensated by other FTs. In addition, we are creating double mutations of ft3b with Gmft5a or Gmft5b, which might help to investigate the fully functions of GmFT3b. Previously, we found that GmFT3b promoted flowering in Arabidopsis. Similarly, both Fan et al. (2014) and Lee et al. (2021) found that ectopic expression of GmFT3b in Arabidopsis can promote the early flowering, suggesting that GmFT3b functions normally as a flowering inducer. Furthermore, in this study, GmFT3b-GFP were expressed in the nucleus of tobacco leaves consistent with other FT proteins in soybean, which implying GmFT3b protein was normally transported within cells.

Revised portions are shown in line 261-262.

References

Fan, C.; Hu, R.; Zhang, X.; Wang, X.; Zhang, W.; Zhang, Q.; Ma, J.; Fu, Y.F. Conserved CO-FT regulons contribute to the photoperiod flowering control in soybean. BMC Plant Biol. 2014, 14, 9.

Lee, S.H.; Choi, C.W.; Park, K.M.; Jung, W.H.; Chun, H.J.; Baek, D.; Cho, H.M.; Jin, B.J.; Park, M.S.; No, D.H., et al. Diversification in functions and expressions of soybean FLOWERING LOCUS T genes fine-tunes seasonal flowering. Front. Plant Sci. 2021, 12, 613675.

Li, X.M.; Fang, C.; Yang, Y.Q.; Lv, T.X.; Su, T.; Chen, L.Y.; Nan, H.Y.; Li, S.C.; Zhao, X.H.; Lu, S.J., et al. Overcoming the genetic compensation response of soybean florigens to improve adaptation and yield at low latitudes. Curr. Biol. 2021, 31, 3755-3767.

Point 2: Figure 7 shows the function of FT3b, but I cannot imagine how FT3b interacts with FT5b, FT2a, and FT5a. What is the function of FT3b in this complex to promote reproductive growth in soybean?

Response 2: Thank you for your careful and constructive comments. We have added an annotation for the Figure 7. Actually, the sizes of boxes represent the effects of FTs in photoperiodic flowering: The bigger the box, the stronger the effect. We supposed that all FTs are involved in regulating the flowering and reproductive growth based on the “weight” model in soybean flowering transition regulation (Chen et al., 2020). GmFT2a and GmFT5a dominate the photoperiodic flowering under the downstream regulatory pathway, while both GmFT5a and GmFT5b may compensate for GmFT3b.

Revised portions are shown in line 314-315.

References

Chen, L.; Cai, Y.P.; Qu, M.G.; Wang, L.W.; Sun, H.B.; Jiang, B.J.; Wu, T.T.; Liu, L.P.; Sun, S.; Wu, C.X., et al. Soybean adaption to high-latitude regions is associated with natural variations of GmFT2b, an ortholog of FLOWERING LOCUS T. Plant Cell Environ. 2020, 43, 934-944.

Reviewer 2 Report

The manuscript entitled “Functional redundancy of FLOWERING LOCUS T 3b in soybean flowering time regulation” by Su et al aimed to study flowering locus homologs (FTs) 3b in soybean and found that GmFT3b is redundant in photoperiodic flowering. 

First, the authors confirmed GmFT3b expression depends on photoperiod and is more responsive to long day (LD) conditions. Second, they investigated the expression level of three GmFT3b overexpression and ft3b knockout soybeans under different photoperiods. The data showed that there were no significant differences in flowering time between them under natural long day, short day, or LD conditions. Therefore, the author concluded that GmFT3b is redundant, and GmFT5a and GmFT5b potentially are compensation for the loss of GmFT3b. Finally, the authors proposed a model to explain the redundancy of FTs regulating flowering in soybean. 

Although the review presents new data and analysis, there are several concerns I have below: 

1.GmFT2a and GmFT5a have been proven to effectively promote flowering through photoperiod regulation. How does the author rule out the possibility that GmFT2a and GmFT5a compensate for the loss of GmFT3b? Did the author check the expression level of GmFT2a and GmFT5a in the ft3b mutant plant? Please give the explanation in the discussion. 

2 Line 223: GCR mechanism can significantly increase expression of other FTs in single ft soybean mutants. Then the authors mentioned that In GmFT3b overexpression soybean plants, no significant differences in FT levels were observed compared with WT plants. The results are very interesting. The author should discuss why overexpression plants have no significant difference in FT levels. 

Author Response

Point 1: GmFT2a and GmFT5a have been proven to effectively promote flowering through photoperiod regulation. How does the author rule out the possibility that GmFT2a and GmFT5a compensate for the loss of GmFT3b? Did the author check the expression level of GmFT2a and GmFT5a in the ft3b mutant plant? Please give the explanation in the discussion.

Response 1: Thank you for your careful and constructive comments. Previously, we showed that ft2aft5a displayed extremely late flowering compared with WT or single mutant plants under LD conditions (Cai et al., 2020). Considering that GmFT3b promoting flowering in Arabidopsis and exhibiting rhythmic changes in soybean, we speculate that GmFT3b is compensated by other major FT genes and is functionally redundant. In figure 6, we showed the expression profiles of GmFT2a, GmFT3a, GmFT5a and GmFT5b in ft3b mutant plants, respectively. As expected, the GmFT5a and GmFT5b were up-regulated in ft3b plants, which might resulted in no phenotype of ft3b plants compared with WT. Furthermore, GmFT5a is confirmed to be the main flowering-promoting factor under LD (Cai et al., 2020; Takeshima, 2019). Thus, we speculate that GmFT5a mainly compensates for the effect of GmFT3b under LD conditions, while GmFT2a and GmFT5b are also involved in compensating the function of GmFT3b. In addition, we are creating double mutations of ft3b with Gmft5a or Gmft5b, which might help to investigate the fully functions of GmFT3b.

Revised portions are shown in line 280-283.

References

Cai, Y.; Wang, L.; Chen, L.; Wu, T.; Liu, L.; Sun, S.; Wu, C.; Yao, W.; Jiang, B.; Yuan, S., et al. Mutagenesis of GmFT2a and GmFT5a mediated by CRISPR/Cas9 contributes for expanding the regional adaptability of soybean. Plant Biotechnol. J. 2020, 18, 298-309.

Takeshima, R.; Nan, H.Y.; Harigai, K.; Dong, L.D.; Zhu, J.H.; Lu, S.J.; Xu, M.L.; Yamagishi, N.; Yoshikawa, N.; Liu, B.H., et al. Functional divergence between soybean FLOWERING LOCUS T orthologues FT2a and FT5a in post-flowering stem growth. J. Exp. Bot. 2019, 70, 3941-3953.

Point 2: Line 223: GCR mechanism can significantly increase expression of other FTs in single ft soybean mutants. Then the authors mentioned that in GmFT3b overexpression soybean plants, no significant differences in FT levels were observed compared with WT plants. The results are very interesting. The author should discuss why overexpression plants have no significant difference in FT levels.

Response 2: Thank you for your careful and constructive comments. We have carefully revised the manuscript according to your comments. GCR mechanism have been reported in various model organisms, such as zebrafish and Arabidopsis thaliana, while the GCR was often observed in gene-knockout mutants but not in gene-knockdown mutants (Bouche et al., 2001; Kok et al., 2015). In soybean, GmFT5a and GmFT2a compensate for each other in their single mutant (Li et al., 2021). Therefore, GmFT5a and GmFT5b might compensate the function of GmFT3b through transcriptional upregulation in ft3b plants. GmFT2a and GmFT5a are potent factors that promote flowering in soybean (Li et al., 2021). Cai et al. (2020) previously showed that soybean plants overexpressing GmFT5a failed to induce early flowering under SD conditions, but could induce early flowering under LD conditions, as only GmFT2a also promoted flowering. Similarly, other FTs were intact in GmFT3b-overexpressed plants, and their expression levels did not change. Thus, we speculated that the expression levels of GmFT2a, GmFT5a and GmFT5b under LD photoperiod were sufficient to induce flowering despite the overexpression of GmFT3b in soybean.

Revised portions are shown in line 269-273.

References

Bouche, N.; Bouchez, D. Arabidopsis gene knockout: phenotypes wanted. Curr. Opin. Plant Biol. 2001, 4, 111-117.

Cai, Y.; Wang, L.; Chen, L.; Wu, T.; Liu, L.; Sun, S.; Wu, C.; Yao, W.; Jiang, B.; Yuan, S., et al. Mutagenesis of GmFT2a and GmFT5a mediated by CRISPR/Cas9 contributes for expanding the regional adaptability of soybean. Plant Biotechnol. J. 2020, 18, 298-309.

Kok, Fatma O.; Shin, M.; Ni, C.-W.; Gupta, A.; Grosse, Ann S.; van Impel, A.; Kirchmaier, Bettina C.; Peterson-Maduro, J.; Kourkoulis, G.; Male, I., et al. Reverse genetic screening reveals poor correlation between morpholino-induced and mutant phenotypes in zebrafish. Dev. Cell 2015, 32, 97-108.

Li, X.M.; Fang, C.; Yang, Y.Q.; Lv, T.X.; Su, T.; Chen, L.Y.; Nan, H.Y.; Li, S.C.; Zhao, X.H.; Lu, S.J., et al. Overcoming the genetic compensation response of soybean florigens to improve adaptation and yield at low latitudes. Curr. Biol. 2021, 31, 3755-3767.

Reviewer 3 Report

In this manuscript, the authors  investigated the possible functional redundancy of the GmFT3b gene. Overall, this study includes extensive experimental results including overexpression and knockout. Also, the manuscript is very well-written and informative. I have two comments/suggestions for the authors to address:

  • Should the functional redundancy of GmFT3b mentioned under certain conditions that the study focuses on instead of a general phenomenon? For example, what if GmFT3b takes role under salinity stress in particular or another environmental condition? Could the authors expand on this.
  • In the phylogenetic tree, why was neighbor joining method selected? Maximum likelihood or bayesian methods could give more accurate tree. 

Author Response

Point 1: Should the functional redundancy of GmFT3b mentioned under certain conditions that the study focuses on instead of a general phenomenon? For example, what if GmFT3b takes role under salinity stress in particular or another environmental condition? Could the authors expand on this.

Response 1: Thank you for your careful and constructive comments. We have carefully revised the manuscript according to your comments. We have been investigating how the FT genes are affected by photoperiod. GmFT3b is a homolog of FT, and its expression is regulated by photoperiod and exhibits rhythmic changes. Therefore, in this study, we focused on the effect of photoperiod on GmFT3b. We planted GmFT3b-related materials in artificial greenhouse (SD and LD) and fields (Beijing, 116°34’E, 39°98’N; Sanya, 109°30’E, 18°18’N) with different photoperiods. But all data revealed that the GmFT3b-related materials did not change flowering times compared with WT although GmFT3b was identified acting as flowering inducer in Arabidopsis (Fan et al., 2014; Lee et al., 2021). It is well known that FT is affected by many factors, such as light, temperature, hormones, and biotic and abiotic stresses (drought, salinity, heat, and nutrient stress). Although we explored the function of GmFT3b under different photoperiods, it would be interesting to study the role of GmFT3b under environmental stress in the future.

Revised portions are shown in line 297-301.

References

Fan, C.; Hu, R.; Zhang, X.; Wang, X.; Zhang, W.; Zhang, Q.; Ma, J.; Fu, Y.F. Conserved CO-FT regulons contribute to the photoperiod flowering control in soybean. BMC Plant Biol. 2014, 14, 9.

Lee, S.H.; Choi, C.W.; Park, K.M.; Jung, W.H.; Chun, H.J.; Baek, D.; Cho, H.M.; Jin, B.J.; Park, M.S.; No, D.H., et al. Diversification in functions and expressions of soybean FLOWERING LOCUS T genes fine-tunes seasonal flowering. Front. Plant Sci. 2021, 12, 613675.

Point 2: In the phylogenetic tree, why was neighbor joining method selected? Maximum likelihood or bayesian methods could give more accurate tree.

Response 2: Thank you for your careful and constructive comments. We have carefully revised the phylogenetic tree according to your comments.

Revised portions are shown in line 117, 331.
